# Evaluation of the Cost-Effectiveness of Evidence-Based Interventions to Increase Female Breast and Cervical Cancer Screens: A Systematic Review

**DOI:** 10.3390/cancers16061134

**Published:** 2024-03-13

**Authors:** Victoria Phillips, Daniela Franco Montoya, E. Kathleen Adams

**Affiliations:** 1Rollins School of Public Health, Emory University, Atlanta, GA 30322, USA; eadam01@sph.emory.edu; 2College of Public Health, University of Georgia, Athens, GA 30602, USA; mbw7@cdc.gov

**Keywords:** evidence-based interventions, screening rates, cost-effectiveness of evidence-based outreach, breast cancer screening, cervical cancer screening

## Abstract

**Simple Summary:**

The National Breast and Cervical Cancer Early Detection Program (NBCCEDP) in the United States supports outreach programs which aim to increase breast and cervical cancer screening rates for low-income, underserved, and uninsured women. Given that resources are limited, we conducted a systematic review of the cost-effectiveness of evidence-based, community outreach programs to increase screening rates among hard-to-reach women. We identified eleven studies for the period 1999–2021, nine for breast cancer and two for cervical cancer. One-on-one education was the most common approach. We found that the average cost to increase breast cancer screening through outreach for one additional woman was USD 545, while the average cost for cervical cancer was USD 197. Cost-effectiveness estimates varied substantially by outreach approach, the population included, and the intervention setting. Uncertainty in cost and effect estimates and program replicability in other settings and with other populations were not addressed, which precludes using existing cost-effectiveness estimates to inform program funding.

**Abstract:**

Purpose: To systematically review published cost-effectiveness analyses of Evidence-Based Interventions (EBIs) recommended by the United States Community Preventive Services Task Force (CPSTF) to increase breast and cervical cancer screening. Methods: We searched PubMed and Embase for prospective cost-effectiveness evaluations of EBIs for breast and cervical cancer screening since 1999. We reviewed studies according to the Consolidated Health Economic Evaluation Reporting Standards (CHEERS) and compared the incremental cost-effectiveness ratio (ICERs), defined as cost per additional woman screened, adjusted to 2021 USD, within and across EBIs by cancer type. Results: We identified eleven studies meeting our review criteria: nine were breast cancer-focused, one breast and cervical cancer combined, and one cervical only, which together reported twenty-four cost-effectiveness assessments of outreach programs spanning eight EBIs. One-on-one education programs were the most common EBI evaluated. The average ICER across breast cancer studies was USD 545 (standard deviation [SD] = USD 729.3), while that for cervical cancer studies was USD 197 (SD = 186.6. Provider reminder/recall systems for women already linked to formal care were the most cost-effective, with an average ICERs of USD 41.3 and USD 10.6 for breast and cervical cancer, respectively. Conclusions: Variability in ICERs across and within EBIs reflect the population studied, the specific EBI, and study settings, and was relatively high. ICER estimate uncertainty and the potential for program replicability in other settings and with other populations were not addressed. Given these limitations, using existing cost-effectiveness estimates to inform program funding allocations is not warranted at this time. Additional research is needed on outreach programs for cervical cancer and those which serve minority populations for either of the female cancer screens.

## 1. Introduction

The American Cancer Society [1,2] estimates that, in 2024, more than 450,000 women will die from breast cancer, and more than 4000 women will die from cervical cancer in the United States. To reduce access barriers, Congress passed the Breast and Cervical Cancer Mortality Prevention Act in 1990 to establish the National Breast and Cervical Cancer Early Detection Program (NBCCEDP) with the goal of increasing screening rates for low-income, uninsured, and underserved women.

Beginning in 1996, the United States (US) Community Preventive Services Task Force (CPSTF) began to identify evidenced-based interventions (EBIs) shown to increase the uptake of preventive cancer screens. Interventions are ranked as recommended, insufficient evidence, and not recommended. This list is updated every three to five years in light of new information from ongoing studies and is often used by outreach programs funded by NBCCEDP to guide intervention choice (See Table 1).

Despite the documented effectiveness of timely breast and cervical cancer screening [3,4], screening uptake remains low. The 2021 US rate for breast cancer screening in the past two years was 75.6% among those 50–75, below the Healthy People 2030 goal of 80.3%. Similarly, the 2021 rate for cervical cancer screening among those 21–65 years old at 73.9%, also below the US goal of 79.2% [5,6]. In addition, 58.8% of low-income and uninsured women, the target population of the NBCCEDP, remained unscreened for breast cancer [7].

Given the limited public health funds for outreach, assessing the cost-effectiveness of EBIs used to increase breast and cervical cancer screening is important, as it can provide information for resource allocation in relation to these and other screening initiatives. Previous reviews have assessed the effectiveness of EBIs for different population groups, cancer types, outcomes, and intervention types. Educational interventions—whether group or individual—have had mixed results.

For rural women in the US, one-on-one education significantly increased breast and cervical cancer screening rates, by close to 20% for the first and more than 10% for the second, while group education was only effective for cervical cancer (Summary OR ≥ 2, 95% CI = 1.3 to 3.6) [8]. In a review focusing on Asian women worldwide, authors found evidence that the effectiveness of education strategies varied with ethnic populations, methods of program delivery, and study settings [9]. When the evaluated outcome was repeated for breast cancer screening, one review and meta-analysis found that education, motivation, or counseling strategies had a lower likelihood of increasing mammography (OR = 1.27, 95% CI = 1.17 to 1.37) than patient reminder strategies (OR = 1.79, 95% CI = 1.41 to 2.29) [10].

Several reviews of EBIs’ cost-effectiveness are available where cost-effectiveness is defined as the cost per additional woman screened based on the ratio of: (Intervention_Cost_ – StatusQuo_Cost_/(Intervention_ScreeningRate_ – StatusQuo_ScreeningRate_) [11]. When the numerator shows a net cost, this is referred to as cost(savings) only. Most reviews provide limited information on the EBI that is evaluated and often rely on micro-simulations, which use data on EBI effectiveness in increasing screening to estimate future costs and survival improvements, with survival impacts expressed in quality-adjusted life-years (QALYs). A recent review of Community Health Worker (CHW) interventions to increase cancer screening, predominantly comprised of microsimulations, found this EBI to be cost-effective for cervical cancer screening when compared to a standardized value of USD 50,000 per QALY. However, the specific activities performed by the CHWs were not detailed [12].

As the use of CHWs has been shown to increase cancer screening rates, they have been incorporated into patient navigation programs [11,12,13,14]. Patient navigation is generally initiated by health care systems and focuses on guiding women who present for screening through the process and any recommended follow-up medical treatment [11]. As our focus is the cost-effectiveness of community outreach, we did not include them in our review.

The cost-effectiveness of multi-component interventions, or those using a combination of EBIs, has also been investigated [15]. The specific EBIs employed were not indicated and studies were classified according to broad strategic categories, such as reducing structural barriers. Studies were not evaluated for methodological rigor. The authors found the median cost per additional woman screened to be USD 197 for breast cancer and USD 159 for cervical cancer [15].

The evaluation methods used to assess the value of the CDC’s National Breast and Cervical Cancer Early Detection Programs (NBCCEDP), for both breast and cervical cancer, have also been evaluated. Authors found a lack of cost-effectiveness research, as only two out of eight studies contained cost-effectiveness analyses. Both studies employed microsimulations and found that NCBCCEDP-funded programs were cost-effective relative to a no-program scenario (ICER values ranged from USD 32,531 to USD 51,754 per QALY) [16].

To our knowledge, there are no reviews assessing the methodological quality of studies evaluating the cost-effectiveness of implementing community-based EBIs. For example, under the NBCCEDP review, neither study took the societal perspective, as recommended, which includes costs outside program delivery [16,17]. No review has examined the cost per additional woman screened between and across EBIs for breast and cervical cancer screening outreach. Our review thus adds to the breast and cervical cancer literature in five ways.

First, we systematically identify EBI cost-effectiveness studies published between 1999 and 2021 using PRISMA guidelines [18]. Second, we assess the identified papers for methodological quality by reviewing their adherence to the Consolidated Health Economic Evaluation Reporting Standards 2022 (CHEERS) [11] (see Section 3). Third, we report the incremental cost-effectiveness ratio (ICER) in 2021 USD, defined as the cost per additional woman screened, by EBI and cancer type, thereby creating a baseline for future between-EBI comparisons. Fourth, we analyze EBI aspects, such as priority populations and implementation settings, which likely affect the cost-effectiveness values that were found. Lastly, we identify methodological weaknesses and areas for future research.

## 2. Materials and Methods

We followed the PRISMA guidelines in identifying studies for inclusion in our literature review [18]. We searched PubMed and Embase for prospective studies implementing EBIs classified by the CPSTF as recommended (strong or sufficient evidence) or insufficient evidence to increase screening rates (See Table 1). We included the latter category to assist in building the CPSTF knowledge base. The search was limited to studies conducted in the US from 1999 to 2021.

We used the following search strategy: (((“cancer”) AND (“breast” OR “cervical”) AND (“screening” OR “mammogram” OR “mammography” OR “pap” OR “mammography screening” OR “pap screening”) AND (“promotion” OR “promote” OR “promoting” OR “increase” OR “Increase uptake” OR “improving” OR “improving uptake”) AND (“cost effectiveness” OR “cost-effectiveness”)) NOT (review[Title])) NOT (Simulation[Title]). We found fewer studies in Embase, and all of them were already included in our PubMed findings. Our inclusion–exclusion process is detailed in Figure 1.

Based on title reviews by two authors per title, we excluded literature reviews (n = 18), analyses for other outcomes (not cervical or breast cancer incentivizing screening programs) (n = 49), and effectiveness evaluations (131). All three authors reviewed abstracts of the remaining records and eliminated studies conducted in non-US settings (n = 11) because of environmental differences which preclude comparisons with US studies and cost-effectiveness microsimulations (n = 4), as they use evidence of EBI effects to model cancer survival instead of directly assessing the impact of EBIs on screening rates. Lastly, we reviewed four papers cited in the references of the papers meeting the inclusion criteria. One was excluded based on author concerns about sample attrition [19].

Except for the one noted, we did not exclude any studies based on sample size as our goal was to include all identifiable EBI implementation studies. Also, implementation trials often have very small samples as they are conducted in small, community-based organizations and their focus is hard-to-reach women.

We report the study characteristics following CHEERS reporting guidelines designed specifically for health economic evaluations [11]. We summarize the key methodological aspects in Section 3 and discuss others in the text. The results were updated from the year of each study’s data to 2021 USD based on the inflation calculator of the Bureau of Labor Statistics [20].

We report the incremental costs, incremental effects, and the incremental cost-effectiveness ratios (ICER) for each study by EBI and cancer type. ICER values are calculated as the cost per additional woman screened, where percent changes in screening are converted to decimals. We present an average ICER for EBIs where multiple cost-effectiveness assessments exist. We rank EBIs in the table from most cost-effective (the lowest ICER) to least cost-effective (the highest ICER) to identify the EBI program where an additional woman can be screened at the lowest cost.

Note that the results in published studies are expressed as percent changes when in fact they are percentage point changes in screening rates after intervention. We follow this convention in our text and tables and report incremental effects as percent changes. Incremental cost-effectiveness ratios (ICERs) are calculated as the cost per additional woman screened, whereby percent changes are converted to decimals. Lastly, consistent with the literature, we defined interventions combining two or more EBIs as bundled or multi-component. 

## 3. Results

### 3.1. Methodological Results

We found 11 papers which met the review criteria: nine targeting breast cancer, two targeting cervical cancer, and one addressing both. Seven papers evaluated multiple outreach programs, resulting in a total of 24 cost-effectiveness EBI assessments: 22 for breast cancer and 2 for cervical cancer [21,22,23,24,25,26,27,28,29,30,31]. Following CHEERS [20], we summarize the characteristics of each study in Table 2 and the cost-effectiveness results in Table 3.

All studies were randomized trials; however, units of randomization varied. Eight of the studies randomized women [19,22,23,24,25,26,27,28] targeted for outreach, one study randomized communities [29], one randomized clinics [30], and one randomized churches [31]. Studies focused on women of different socioeconomic backgrounds with an emphasis on reaching low-income [23,27,28,31], rural [28,29], and minority women [28,31]. Other priority populations included urban women [27], uninsured women [30], and veterans [24]. Four studies did not name a specific population [19,22,25,26].

Evaluations were conducted in community settings [23,28,30,31], clinical settings [19,24,27,29], and managed care settings [22,25,26]. The three studies in managed care settings took a healthcare provider perspective [22,25,26], and three of the community-based evaluations took a program perspective [23,28,30], while four took the societal perspective, per recommended guidelines, and accounted for the average participant’s time spent in the intervention [24,27,30,31].

Eight studies reported the time associated with the specific outreach activity [22,23,24,25,26,27,29,30], with participants spending an average of 14.8 min in an intervention, with the shortest intervention being a three-minute phone counseling session [29] and the longest intervention a 64 min in-person counseling session, plus a letter reminder [25]. The average analytic horizon, the time used to confirm whether a woman was screened following outreach across studies, was 1.31 years (SD = 0.73).

Researchers took a micro-costing approach, whereby the resources used to implement the intervention identified, tracked, and assigned a unit cost. Then, total costs and average costs per participant were calculated. While all studies tracked labor and materials costs, ongoing facility operating costs were only included in two studies developed in health care settings [22,27]. Three studies accounted for fixed costs [24,26,31]. Only one study included recruiting and training costs [29]. Two studies valued patient’s time at the minimum wage [24,30], and two studies assumed a wage value [26,31]. Only two studies accounted for the cost of regular care for the control group, while others assumed a baseline cost of zero [19,22]. Screening costs were not included in any of the papers reviewed. All studies assumed a baseline effectiveness equal to the average screening rate for the control group, except the paper by Andersen et al., which assumed a baseline effectiveness of zero [29].

For breast cancer, eight EBIs were evaluated. One-on-one education was the most common, with eight assessments being conducted across five studies [22,23,25,29,31], followed by client reminders, with five assessments across five studies [21,23,24,25,26]. One study evaluated a provider education strategy, which we categorized as a provider incentive EBI. Six programs using bundled or multi-component EBIs: four combined one-on-one education and client reminders [23,25,26], two one-on-one education and group education [28], and one-on-one education with provider reminder/recall system plus one-on-one education, client reminders and measures to reduce structural barriers [27]. Two EBIs, with one assessment each, were evaluated for cervical cancer: one-on-one education and a provider remined/recall system. All EBIs were recommended, bar provider incentives with insufficient evidence.

### 3.2. Cost-Effectiveness Analysis

Table 3 summarizes the cost-effectiveness study results by EBI and cancer type. EBIs are ranked by the incremental cost-effectiveness ratio, from low to high, by cancer type. Global values and median values for costs incremental costs, screening rates, changes in screenings, and the ICERs are shown at the bottom of the table.

### 3.3. Breast Cancer Screening Programs

Incremental costs. The average global incremental cost per participant across breast cancer studies was USD 24.7 (SD = USD 38.5). The lowest incremental cost per participant across EBIs (USD 0.8) was a client reminder strategy combined with a letter plus automated calls, while the highest (USD 2851.6) was the provider incentive strategy of a physician-based education program [19,22].

Incremental effects. The average increase in screening rates across EBIs was 7.7% (SD = 7.32%). One-on-one education was the least effective, with five of the eight outreach programs assessed falling below the global average effectiveness. The most effective EBI, increasing screening rates by 27%, was a bundled program which included provider reminders and recall systems, one-on-one education, client reminders, and the removal of structural barriers [27].

Incremental cost-effectiveness ratios (ICERs). The average global ICER across breast cancer studies was USD 545.1 (SD = USD 729.3) with substantial variation around the value. The EBI of provider reminders and recall systems, represented by a single study of automated reminders, proved to be the most cost-effective, primarily due to its low cost, even though its effectiveness, at 4.6%, was lower than the global average [27].

The combination of one-on-one education + client reminders, evaluated in two programs, had the second lowest average ICER at USD 105.3, (SD = USD 84.1) across breast cancer EBIs. Variation was significant, as both the minimum ICER (USD 38.7) and the maximum ICER (USD 244.4) were associated with a scheme of telephone counseling sessions plus mailed reminders [23,26]. The difference between the two reflects that the lowest cost intervention used the Managed Care Organization’s computerized tailored algorithm to identify women for mailings; the highest cost intervention was community-based and used third-party-collected data.

Client Reminders ranked third in terms of cost-effectiveness, at USD 309.4 (SD = ISD 415.2) based on evaluations of five programs. The two physician reminder interventions resulted in no increase in the number of women screened [24,25]. A combination of letter and phone reminders resulted in the lowest ICER (USD 4.5) [19], followed by an intervention of only letter reminders tailored to patient’s needs and health history (USD 27.4) [19,26]. Both had relatively large increases in screening rates, at 17.7% and 10.6%, respectively, and a very low average cost per participant, at USD 0.8 and USD 2.9, respectively.

The most comprehensive bundled outreach program comprised provider reminders, one-on-one education, client reminders, and a reduction in structural barriers [27]. It resulted in the greatest increase in mammography screening at 27%, but ranked fourth in terms of ICERs due to its relatively high cost.

One-on-One Education, the most commonly implemented EBI, ranked fifth, with an average ICER of USD 421.9 (SD: USD 459.1). Most studies in this category were low-cost, but their effectiveness was also low, with five below the average global effectiveness of a 7.7% increase in screening rates [23,25,29,31]. In-person counseling, delivered by nurses during clinic appointments, demonstrated the highest effectiveness, with a 15.9% increase in screening rate [25].

The three least cost-effective EBIs, all examined in single programs, consisting of group education, one-on-one + group education, and provider incentives, were at least double the global average ICER. For example, one-on-one education + group education, ranked seventh, had a relatively high ICER at USD 1625, compared to the global average, and low effectiveness, with only a 2% increase in screening rate. The one-on-one education component in this intervention was delivered telephonically, while the group education component was led, in-person, by CHWs, thus making the intervention labor intensive and costly [29].

### 3.4. Cervical Cancer Screening Programs

Only two studies evaluated interventions to promote cervical cancer screening, with the average ICER across them being USD 197 (SD = 186.6). The incremental cost per participant of the provider reminder and recall system, at USD 1.5, was far lower than that for the one-on-one counseling strategy, at USD 74.5 [28,30]. While the one-on-one education strategy for cervical cancer was more effective than a provider reminder with a recall system, at 19.4% compared to 15.0%, the higher cost per participant led to it having a higher ICER.

## 4. Discussion

Summary. Groups that have been marginalized face substantial challenges in accessing breast and cervical cancer screening. The NBCCEDP aims to increase screening among women with low-incomes and those who lack insurance by providing screening and diagnostic services and supporting the implementation of CPSTF-designated EBIs, which aim to increase screening rates among hard-to-reach women. Given the limited public health resources and multiple EBI options, documenting the cost-effectiveness of specific EBIs is informative.

The average cost per additional woman screened for breast cancer showed substantial variation, at USD 545.1 (SD = USD 729.3) with differences in EBI type, study population, setting, delivery methods, and specific resource costs precluding direct comparisons of the approaches. Given this, comparisons of the relative cost-effectiveness between EBIs should be undertaken with caution.

Incremental costs varied from USD 0.80 to USD 171, while incremental effectiveness ranged from 1.8 to 27% with two reminder programs producing no increase in screening uptake. In terms of high ICER values, low effectiveness was the key driver. We observed that the costliest strategies (USD 171.15 [31] and USD 100.25 [25]) had similar costs, but substantially different effects (6% vs. 27%) which resulted in notably different ICERs values (USD 2852.42 and USD 371.3) [25,31]. The lack of cervical cancer research precludes EBI comparisons.

Consistent with previous reviews [9], we found that education and counseling interventions alone have relatively low effectiveness, with a 5.6% average increase in screening rates. These interventions were also often labor-intensive, leading to relatively high costs. Low-labor-intensity strategies—such as letter reminders—had a relatively low impact on screening rates among non-compliant women, but they were highly effective when they reached women already linked with some form of formal care, such as those with a history of mammograms [26]. Taken together, these results suggest that the intervention’s labor intensity does not consistently correlate with effectiveness.

Efforts to link women reached through community-based outreach with clinical partners remain an important area of future research. Four EBIs that were implemented were initiated from clinical providers which leveraged existing databases to increase screening, which proved relatively effective. [19,23,27,29]. Out of the four community-based studies, two specified clinical sites to which women were referred for screening, suggesting a strong community–clinical linkage [23,28]. Establishing explicit relationships between community entities and clinical providers may increase screening uptake, as this may reduce access barriers.

Other approaches to increasing screening also warrant consideration. Web-based and social media applications (App) are at the initial investigation stage in terms of outreach [36,37]. Exploring outreach models to increase, for example, HIV testing, may be information. An App which identified lesbian, gay, bi-sexual, transgender (LGBT)-friendly HIV testing sites proved to be effective in increasing the uptake of HIV testing and was likely not cost-prohibitive for medium-sized public health departments [38].

Apps which provide location information can help minimize travel costs, can be tailored to populations with different needs, such as providing translation assistance, and can provide notifications, for example, on the availability of mobile vans. Exploring the use of interactive Chatbots in relation to cancer care is also being explored [39]. A recent review provided information on the engagement and reach of different social media approaches for increasing the uptake of HPV vaccination and may provide insight into models for breast and cervical cancer screening [40].

Bundled interventions showed mixed results in terms of cost-effectiveness. One-on-one education coupled with group education ranked seventh in terms of relative cost-effectiveness, while a bundled intervention drawing on existing clinical information was extremely effective. Adding components increases costs, and thus additional elements should be considered carefully. Recent work has shown that simultaneously targeting multiple cancers, rather than increasing the number of outreach elements, may improve the cost-effectiveness of outreach programs [37].

Our findings suggest that multi-component strategies, including participation by CHWs, produced the best results for cervical cancer screening among low-income women (19.41% increase) in rural areas [30]. However, the ICER estimates in our review are higher than those reported previously for multi-component strategies [15].

Most studies (three out of four) targeting rural and low-income women using CHWs had effectiveness values below the global effectiveness average, thus reflecting the challenges of increasing breast cancer screening among women who are hard to reach and may face distance barriers [23,29,31]. For example, targeting women drawn from an existing clinical database had lower resource requirements and a higher likelihood of increasing screening rates than community outreach programs targeting uninsured or hard-to-reach populations.

Low effectiveness generates high ICER values, other things being equal, and may discourage investment in outreach programs even though, arguably, these are directed at some of the women in greatest need. Given this, cost-effectiveness comparisons alone should not be the criterion on which the value of these studies should be assessed.

The lack of studies focusing on EBIs for cervical cancer screening is concerning due to the relatively low screening rates among women in need and considering that pre-cancerous lesions can be identified, and further progression prevented, through screening [32,33,41], while incident cases can be prevented by HPV vaccination [34]. A future investigation of interventions to increase cervical cancer screening, noting those which offer the greatest value, is needed.

Study Quality. In terms of methodological quality, all studies in our review used a micro-costing approach and generally followed the CHEERS guidelines [20], with two significant differences. Only two studies accounted for fixed costs. Saywell et al. [25,26] argued that their inclusion is often inconsequential, as they often represent a very small part of the total expenses. Our results support this claim, as we found that fixed costs constituted less than 3% of the total costs reported in the study. However, consideration of the upfront training and/or recruitment of outreach workers is warranted, particularly as this may prove a challenge for small, community-based programs and those in rural settings with limited budgets.

Similarly, we found that seven of the eleven studies did not address participant costs. This omission is unlikely to bias ICER values as most strategies took little time, comprising less than three percent (about 2.4%) of the overall costs. However, assessing participant costs beyond intervention time is important. For example, travel time, transportation, and securing childcare may impose additional costs on participants and act as barriers to screening. Some programs specifically target eliminating transportation costs through travel vouchers [27].

Other concerns exist across effect measures. Most studies relied on the unverified self-report of outcomes. While self-reporting has been found to be accurate in population surveys regarding mammography and Pap smear after correcting for overreporting [35], no study investigated this effect. Furthermore, issues related to missing data were often not addressed, and no sensitivity analyses were addressed in any of the studies. Sensitivity analyses were conducted in a single paper, and all would benefit from examining the effect of possible variations in key values on their results. [23].

## 5. Limitations

There are several limitations to this review. First, we were unable to provide a statement about overall EBI or specific EBI cost-effectiveness relative to a QALY-based standard as we focused on prospective outreach research that followed participants only to the point of screening. Furthermore, we did not analyze studies of the cost-effectiveness of follow-up or repeated cancer screening. Our findings are also limited to outreach strategies for initial patient–provider linkages, not clinical outcomes, or screening compliance. They also focus only on the US. Lastly, the lack of cost-effectiveness research for cervical screening outreach precluded conclusions for this type of cancer. This is a limitation that affects any review of this type.

We were not eligible for registration in PROSPERO as we began data extraction prior to our attempt to register. We do not believe this introduced any bias in terms of the studies identified, given the open structure of our search criteria, nor do we believe that it negatively impacted the study methodology [42]. Using the CHEERS recommendations, we assessed the quality of reporting of the studies, rather than the quality of their conduct.

This review differs from several others in that we assess identified studies and strategies as EBIs, focusing on their methodological quality, and their reporting of cost per additional woman screened by EBI and cancer type. We thus provide a baseline for further between-EBI comparisons, which serves as a starting point for future research.

However, the NBCCEDP is increasingly interested in considering cost-effectiveness in allocating outreach support dollars. Our findings underscore that caution should be exercised, given the current state of the literature. Additional investigation is needed to determine if sufficient evidence for the intervention and provider incentives supports full recommendation. Critically, ICER estimates are highly variable and important methodological issues remain unaddressed. A potentially important area for NBCCEDP to fund is more robust cost-effectiveness, including increased sample sizes with a focus on replicability.

In addition, cost-effectiveness analyses using microsimulations are dependent on the initial effect of outreach activities, which ultimately determines the number of women screened and, thus, the resulting benefits in terms of longevity and improved clinical outcomes. The variation in the effectiveness of outreach should be addressed in such modelling efforts.

Also, while the NBCCEDP aims to increase screening rates among hard-to-reach women, the cost-effectiveness literature has not generally focused on minorities within this group, as only one study, for example, focused on African American women.

## 6. Conclusions

Given that resources for female cancer screening are limited, comparing the value of alternative outreach approaches is warranted. We conducted a systematic review of the cost-effectiveness of evidence-based, community outreach programs to increase cancer screening rates among underserved women. Based on papers meeting our search criteria, we found that the average ICER for breast cancer screening was estimated to be USD 545 and that for cervical cancer, estimated to be USD 197. Cost-effectiveness estimates varied substantially within and across EBIs, the population served, and the intervention setting. The inherent uncertainty in cost and effect estimates and the potential for program replicability in other settings and with other populations were not addressed. Given these limitations, use of existing cost-effectiveness estimates to inform program funding allocations is not warranted at this time. In addition to addressing these shortcomings, future work needs to focus on evaluations of outreach programs to increase screening rates for cervical cancer and those which serve minority populations for either of the female cancer screens. 

## Figures and Tables

**Figure 1 cancers-16-01134-f001:**
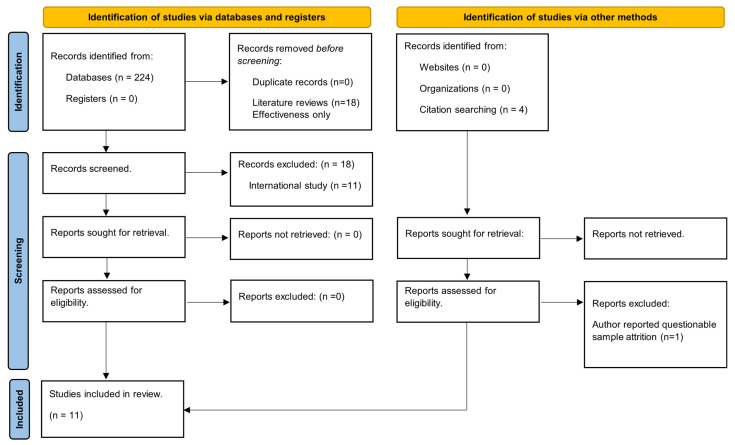
PRISM flow diagram [18].

**Table 1 cancers-16-01134-t001:** Description of evidenced-based interventions and strength of evidence supporting their use.

EBI	Description ^1^	Breast Cancer	Cervical Cancer
Community Health workers	Interventions that engage frontline health workers who serve as a bridge between communities and healthcare systems.	Recommended	Recommended
Client Incentives	Small, non-coercive rewards aim to motivate people to seek cancer screening for themselves or others.	Insufficient evidence	Insufficient evidence
Client Reminders	Written or telephone messages (including automated messages) advising people that they are due for screening.	Recommended	Recommended
Group Education	Group education conveys information on indications for, benefits of, and ways to overcome barriers to screening to inform and encourage participants to seek recommended screening.	Recommended	Insufficient evidence
One-on-One Education	One-on-one education delivers information to individuals about indications for, benefits of, and ways to overcome barriers to cancer screening to inform and encourage to seek recommended screening.	Recommended	Insufficient evidence
Mass Media	Television, radio, newspapers, magazines, and billboards used to communicate educational and motivational information about cancer screening.	Insufficient evidence	Insufficient evidence
Reducing Client Out-of-Pocket Costs	Interventions that attempt to eliminate or minimize economic barriers	Recommended	Insufficient evidence
Reducing Structural Barriers	Interventions designed to reduce obstacles to people’s access to cancer screening.	Recommended	Insufficient evidence
Small Media	Videos and printed materials such as letters, brochures, and newsletters.	Recommended	Recommended
Provider Incentives	Direct or indirect rewards to motivate providers to perform cancer screenings or make an appropriate referral for their patients to receive these services.	Insufficient evidence	Insufficient evidence
Provider Reminder and Recall Systems	Reminders inform health care providers it is time for a client’s cancer screening test.	Recommended	Recommended

^1^ Taken from the community guide CPSTF Findings for Cancer Prevention and Control: https://www.thecommunityguide.org/content/task-force-findings-cancer-prevention-and-control (accessed on 1 August 2022).

**Table 2 cancers-16-01134-t002:** Methodological review of studies by Consolidated Health Economic Evaluation Reporting Standards (CHEERS).

#	Study	Study Characteristics	Cost and Effectiveness Estimation	
[29]	(Andersen et al., 2002)	Screening: Mammography Setting: Community. Perspective: Program. Intervention: Individual counseling (ICs), Community activities (CAs), and a combined intervention (ICCAs). Target population: Rural women. Average Participant Time: 13.9 min. Analytic horizon ^1^: 2.5 years. EBI: CHWs, Group Education, One-on-One Education.	Costing method: Micro. Quantity cost source: Intervention materials records, volunteers time records, personnel time records, estimates of patient time spent on intervention based on final follow-up survey. Value cost source: Unit costs of materials were obtained from invoices. Volunteer and patient costs were calculated based on the salary of a program assistant or a field research coordinator depending on the skill level required for the task. Screening costs: No.	Cost items included: Recruitment, training, labor, materials. Fixed costs: No. Baseline Effectiveness ^2^ = 0. Baseline cost ^3^ = 0.
[30]	(Chirikos et al., 2004)	Screening: Mammography, Pap. Setting: Primary care clinics. Perspective: Payer, Participant. Intervention: Cancer Screening Office Systems (SOS) of chart reminders for clinicians. Target population: Providers of a county-funded health insurance plan for uninsured women. Average Physician Time: 5.12 min. Analytic horizon: 1 year. EBI: Provider Reminder and Recall Systems.	Costing method: Micro. Quantity cost source: Intervention materials records, personnel time records, average patient’s time spent on survey. Value cost source: Mean hourly wage Bureau of Labor Statistics for personnel, and minimum hourly wage for patient’s time. Screening costs: No.	Cost items included: Labor, materials. Fixed costs: No. Baseline Effectiveness ≠ 0. Baseline cost = 0. Baseline effectiveness source: Clinic records baseline control group.
[22]	(Costanza et al., 2000)	Screening: Mammography. Setting: HMO. Perspective: Healthcare provider. Intervention: Barrier-specific telephone counseling (BSTC) and a physician-based educational intervention (MD-ED). Target population: HMO members Average Participant Time: 5.5 min. Analytic horizon: 2 years. EBI: Client Reminders, Provider Reminder and Recall Systems, One-on-One Education.	Costing method: Micro. Quantity cost source: Intervention materials records, personnel time records. Value cost source: HMO records. Screening costs: No.	Cost items included: Labor, materials, facility. Fixed costs: No. Baseline Effectiveness ≠ 0. Baseline cost ≠ 0. Baseline cost source: Reminder control group.
[23]	(Crane et al., 2000)	Screening: Mammography. Setting: Community. Perspective: Program. Intervention: Multiple outcalls, single outcall, advance card + single outcall. Target population: Low-income women. Average Participant Time: 17.6 min. Analytic horizon: 2 years. EBI: One-on-One Education, Client Reminders.	Costing method: Micro Cost source: Intervention records. Quantity cost source: Intervention materials records, personnel time records. Value cost source: Printing and postage costs were actual per item costs, personnel costs used the national average hourly wage of CIS telephone information specialists in 1994 plus a fringe benefit rate of 26% and overhead/direct cost rate of 45%. Screening costs: No.	Cost items included: Labor, materials. Fixed costs: No. Baseline Effectiveness ≠ 0. Baseline cost = 0. Baseline efficiency source: Authors’ assumptions of two possible scenarios.
[24]	(Lairson et al., 2011)	Screening: Mammography. Setting: Veteran Health Administration. Perspective: Payer, Participant. Intervention: Two intervention groups that varied in the extent of mail reminders personalization (tailored vs. targeted). Target population: U.S. Veterans. Average Participant Time: 15.4 min. Analytic horizon: 2 years. EBI: Client Reminders.	Costing method: Micro. Quantity cost source: Intervention materials records, personnel time records, average patient’s time spent in survey. Value cost source: Materials were valued at market prices, wage rates and benefit levels obtained from the Veteran affairs data adjusted to 2008 US dollars by the CPI, patient’s time was valued at their working wage, or the federal minimum wage of 2005 if they were not working. Indirect costs were obtained by multiplying direct costs by an indirect rate of 30%. Screening costs: No.	Cost items included: Labor, materials, facility. Fixed costs: Yes. Baseline Effectiveness ≠ 0. Baseline cost = 0. Baseline efficiency source: Survey only control identified through the Department of Veterans Affairs.
[21]	(Phillips et al., 2015)	Screening: Mammogram. Setting: Suburban family medicine practice. Perspective: Payer. Intervention: Personalized mailed letters, automated telephone calls, or both. Target population: Suburban family medicine practice patients. Average Participant Time: Undetermined. Analytic horizon: 0.29 years. EBI: Client Reminders.	Costing method: Micro. Quantity cost source: Intervention materials records, personnel time records. Value cost source: Unclear. Screening costs: No.	Cost items included: Labor, materials. Fixed costs: No. Baseline Effectiveness ≠ 0. Baseline cost ≠ 0. Baseline cost source: Letter only intervention costs. Baseline efficiency source: Survey personalized letter group. Participants were identified using the family medicine practice records system.
[25]	(Saywell Jr. et al., 1999)	Screening: Mammography. Setting: HMO. Perspective: Healthcare provider. Intervention: 5 combinations of physician recommendation letters and telephone or in-person individualized counseling strategies. Target population: HMO members. Average Participant Time: 33 min. Analytic horizon: 1.3 years. EBI: Client Reminders, One-on-One Education.	Costing method: Micro. Quantity cost source: Intervention materials records, personnel time records. Value cost source: Intervention materials records, personnel time records. Screening costs: No.	Cost items included: Labor, materials. Fixed costs: No ^4^ Baseline data: Yes. Effectiveness: Increase in mammography rate. Baseline Effectiveness ≠ 0. Baseline cost = 0. Baseline efficiency source: Survey no counseling or letter control group identified through medical records from a large HMO and a general medicine clinic.
[26]	(Saywell Jr. et al., 2004)	Screening: Mammography. Setting: MCOs and General medicine clinic. Perspective: Healthcare provider. Intervention: Tailored telephone, tailored physician letter, tailored telephone + tailored physician letter. Target population: MCO members and General medicine clinic patients. Average Participant Time: 13 min. Analytic horizon: 1.3 years. EBI: Client Reminders, One-on-One Education.	Costing method: Micro. Quantity cost source: Intervention materials records, personnel time records. Value cost source: Intervention materials records, personnel time records. Screening costs: No.	Cost items included: Labor, materials. Fixed costs: No ^4^. Baseline data: Yes. Effectiveness: Increase in mammography rate. Baseline Effectiveness ≠ 0. Baseline cost = 0. Baseline efficiency source: Survey control group identified through computer lists from a hospital’s general medicine clinic and two managed care organizations.
[31]	(Stockdale, et al., 2000)	Screening: Mammography. Setting: Community. Perspective: Program, Patient. Intervention: Telephone counseling (Church-based). Target population: Low income, minority. Average Participant Time: Undetermined Analytic horizon: 0.75 years. EBI: CHWs, One-on-One Education.	Costing method: Micro. Quantity cost source: Intervention materials records, personnel time records. Value cost source: Adjusted salary per minute is the base salary level and fringe benefits, which are calculated at 22% of salary for church personnel [32] and 30% for patients [33]. Overhead costs were 25% of total direct costs (including volunteers’ times valued at the minimum wage). Cost of reproducing materials for telephone and mailed components of the intervention were based on the prevailing prices of reprographic services at a local commercial copy center. Screening costs: No.	Cost items included: Labor, materials, facility. Fixed costs: Yes. Baseline Effectiveness ≠ 0. Baseline cost = 0. Baseline efficiency source: Survey control group.
[27]	(Thompson, et al., 2002)	Screening: Mammography. Setting: Public hospital. Perspective: Payer, Participant. Intervention: Program emphasized nursing involvement included physician education, provider prompts, use of audiovisual and printed patient education materials, transportation assistance in the form of bus passes, pre-appointment telephone or postcard reminders, and rescheduling assistance. Target population: Low income, urban. Average Participant Time: 14.9 min. Analytic horizon: 1.17 years. EBI: Provider Reminder and Recall Systems, One-on-One Education, Client Reminders, Reducing Structural Barriers.	Costing method: Micro. Quantity cost source: Intervention materials records, personnel time records, miscellaneous and overhead costs were estimated through hospital historical accounting data. Value cost source: Overhead costs were estimated through hospital historical accounting data, salaries were based on the staff position (nurse, physician, etc.). Overhead costs were assumed to be 28.7% from a range of 24 to 37% based on the previous literature [34,35]. Patient’s opportunity costs were valued based on a USD 25.000 salary per year (0.20/min). Screening costs: No.	Cost items included: Labor, materials, facility. Fixed costs: Yes. Baseline Effectiveness ≠ 0. Baseline cost = 0. Baseline efficiency source: Survey control group. Patients were tracked via clinic records.
[28]	(Thompson, et al., 2017)	Screening: Pap. Setting: Rural Community. Perspective: Program. Intervention: low-intensity (video), high-intensity (video + home-based educational session). Both lead by CHWs called Promotoras. Target population: Low-income, rural. Average Participant Time: Undetermined. Evaluation timeline: 0.083 years. EBI: CHWs, One-on-One Education, Client Reminders.	Costing method: Micro. Quantity cost source: Intervention materials record, Personnel time records. Indirect costs were calculated as recommended by [32]. Value cost source: Unclear. Screening costs: No.	Cost items included: Labor, materials. Fixed costs: No. Baseline Effectiveness ≠ 0. Baseline cost = 0. Baseline efficiency source: Survey control group. Patients tracked via Yakima Valley Farm Workers Clinic (YVFWC).

^1^ The analytic horizon is the period used to determine whether women were screened as a result of the intervention ^2^ Baseline effectiveness is the effectiveness of the status quo or screening rates prior to the intervention. ^3^ Baseline casts are costs associated with the status quo of outreach prior to the intervention. ^4^ The author acknowledged fixed costs, but assumed they were approximately zero for the evaluated intervention.

**Table 3 cancers-16-01134-t003:** Costs, screening rates and cost per additional women screened by evidence-based intervention and cancer type.

EBI	Author	Baseline Cost 2021 Dollars	Average Cost Per Participant 2021 Dollars	Incremental Cost 2021 Dollars	Screening Rate at Baseline	Screening Rate Post Intervention	Percent Change in Screening Rate ^1^	Cost Per Additional Women Screened in 2021 Dollars ^2^	ICER Group Average and Standard Deviation ^3^	Overall ICER Ranking Across EBI Group
Breast cancer screening										
Provider Reminder and Recall Systems	[30] (Chirikos et al., 2004)	USD 0.0	USD 1.9	USD 1.9	71.1	75.67	4.6	USD 41.3	USD 41.3	1
One-on-One Education+ Client Reminders	[26] (Saywell Jr. et al., 2004)	USD 0.0	USD 6.5	USD 6.5	32.6	49.38	16.8	USD 38.7	USD 105.3 (USD 84.1)	2
	[25] (Saywell Jr. et al., 1999)	USD 0.0	USD 8.3	USD 8.3	18.2	35.6	17.4	USD 47.7		
	[25] (Saywell Jr. et al., 1999)	USD 0.0	USD 11.1	USD 11.1	18.2	30.5	12.3	USD 90.24		
	[23] (Crane et al., 2000)	USD 0.0	USD 4.4	USD 4.4	0.0	1.76	1.8	USD 244.4		
Client Reminders	[24] (Lairson et al., 2011)	USD 19.7	USD 41.5	USD 21.8	46.9	46	−0.9	Dominated	USD 309.4 (USD 415.2)	3
	[25] (Saywell Jr et al., 1999)	USD 0.0	USD 0.8	USD 0.8	18.2	15	−3.2	Dominated		
	[21] (Phillips et al., 2015))	USD 2.1	USD 2.9	USD 0.8	18.9	36.6	17.7	USD 4.5		
	[26] (Saywell Jr et al., 2004)	USD 0.0	USD 2.9	USD 2.9	32.6	43.27	10.6	USD 27.4		
	[24] (Lairson et al., 2011)	USD 0.0	USD 19.7	USD 19.7	44.7	46.9	2.2	USD 896.4		
Provider Reminder and Recall Systems, One-on-One Education, Client Reminders, Reducing Structural Barriers	[27] (Thompson et al., 2002)	USD 0.0	USD 100.3	USD 100.3	22.0	49	27.0	USD 371.5	USD 371.5	4
One-on-One Education	[26] (Saywell Jr. et al., 2004)	USD 0.0	USD 3.3	USD 3.3	32.6	41.91	9.3	USD 35.5	USD 421.9 (USD 459.1)	5
	[25] (Saywell Jr. et al., 1999)	USD 0.0	USD 8.7	USD 8.7	18.2	34.1	15.9	USD 54.7		
	[23] (Crane et al., 2000)	USD 0.0	USD 6.4	USD 6.4	0.0	6.56	6.6	USD 97.7		
	[25] (Saywell Jr. et al., 1999)	USD 0.0	USD 7.7	USD 7.7	18.2	23.1	4.9	USD 157.1		
	[23] (Crane et al., 2000) ^4^	USD 0.0	USD 3.7	USD 3.7	0.0	2.0	2.0	USD 185		
	[22] (Costanza et al., 2000)	USD 21.7	USD 65.8	USD 44.0	38.0	47	9.0	USD 489		
	[31] (Stockdale et al., 2000)	USD 0.0	USD 33.2	USD 33.2	0.0	3.24	3.2	USD 1037.5		
	[29] (Andersen et al., 2002)	USD 0.0	USD 21.1	USD 21.1	0.0	1.6	1.6	USD 1318.75		
Group Education	[29] (Andersen et al., 2002)	USD 0.0	USD 32.4	USD 32.4	0.0	2.5	2.5	USD 1296	USD 1296	6
One-on-One Education + Group Education	[29] (Andersen et al., 2002)	USD 0.0	USD 32.5	USD 32.5	0.0	2.0	2.0	USD 1625	USD 1625	7
Provider Incentives (Insufficient Evidence)	[22] (Costanza et al., 2000)	USD 21.7	USD 192.9	USD 171.1	38.0	44	6.0	USD 2851.6	USD 2851.6	8
Global Average		USD 3.0 (USD 7.2)	USD 27.6 (USD 43.19)	USD 24.7 (USD 38.66)	21.3 (18.99)	29.0 (21.00)	7.7 (7.32)	USD 545.1 (USD 729.5)	-	-
Global Median		USD 0.0	USD 8.6	USD 8.6	18.2	34.9	5.5	USD 170.2	-	-
Cervical cancer screening										
Provider Reminder and Recall Systems (Pap)	[30] (Chirikos et al., 2004)	USD 0.0	USD 1.5	USD 1.5	48.2	62.4	14.2	USD 10.6	USD 10.6	1
One-on-One Education	[28] (Thompson et al., 2017)	USD 0.0	USD 74.5	USD 74.5	34.0	53.4	19.4	USD 384	USD 384	2

^1^ Percent change in screening rate = Percentage point change. ^2^ ICER denominator expressed in decimal format. ^3^ Average excludes dominated interventions. ^4^ ICER would have to be divided by the count of women screened after the intervention to reflect the author’s calculations.

## Data Availability

All data used are drawn from the literature and thus freely available.

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
