# Peer review of "Evaluation of the Cost-Effectiveness of Evidence-Based Interventions to Increase Female Breast and Cervical Cancer Screens: A Systematic Review"

_cancers, 2024, doi:10.3390/cancers16061134_

Round 1
Reviewer 1 Report
Comments and Suggestions for Authors
The authors present an interesting manuscript to evaluate the cost-effectiveness of evidence-based interventions aimed at improving adherence to screening programs. This manuscript could be useful in the implementation of different strategies.
However, there are major concerns mainly related to the methodology section.
Some interventions are classified as recommended and other as insufficient evidence. Authors should discuss the rationale of including both types of recommendations in the same review. Additionally, a deeper exploration of the results stratified by this classification would enhance the clarity and relevance of the findings.
The methodology does not follow PRISMA recommendations and thus, the authors should not state that it is a systematic review:
- Authors should provide a detailed account of the search strategy, potentially as a supplementary table, is necessary.
- Is the systematic review in PROSPERO? If not, discuss the limitations of this in the discussion section.
- Figure 1 should adhere to the PRISMA flow diagram format, accurately depicting the articles retrieved from each database in distinct boxes, with the inclusion of the number of duplicated articles. Furthermore, the exclusion of 199 articles based solely on title screening necessitates careful consideration, and the abstracts of these studies should undergo thorough evaluation.
- Detailed inclusion and exclusion criteria, including factors such as sample size considerations, must be clearly outlined to enhance transparency and reproducibility. Additionally, clarification regarding the independence of article selection, variable extraction, and methodological quality assessment by two reviewers is imperative.
Finally, the discussion section should provide a comprehensive justification of the practical implications of the study's findings. Without this, the manuscript risks being perceived as a mere summary of existing literature rather than a meaningful contribution to the field.
Author Response
Reviewer #1:
The authors present an interesting manuscript to evaluate the cost-effectiveness of evidence-based interventions aimed at improving adherence to screening programs. This manuscript could be useful in the implementation of different strategies.
However, there are major concerns mainly related to the methodology section.
Some interventions are classified as recommended and other as insufficient evidence. Authors should discuss the rationale of including both types of recommendations in the same review. Additionally, a deeper exploration of the results stratified by this classification would enhance the clarity and relevance of the findings.
Response: We agree this needs further clarification. We included EBIs which were either recommended by the CPSTF or noted as having insufficient evidence to increase screening rates (See Table 1). We included the latter category to assist in building the CPSTF knowledge base and to provide the widest review of the literature possible. We found one EBI which fell into the “insufficient evidence” category and have added text to emphasize this fact and its implications.
The methodology does not follow PRISMA recommendations and thus, the authors should not state that it is a systematic review:
We respectively disagree with the reviewer. We did follow PRISMA guidelines, but agree that we did not provide sufficient detail in our paper to make this clear, particularly in relation to the reproducibility of our initial literature search. We have now included specificity about the search criteria, the inclusion/exclusion criteria, and the selection process through an updated Figure and additional text. We also re-reviewed the PRISMA checklist and provided additional detail on relevant items, such as 8, 9, 11, 13b. Also, the papers meeting the review criteria themselves used rudimentary statistical methods and several had small samples as the interventions were implemented by community-based organization. Thus, we did not attempt to conduct further analysis on the results, such as subgroup analysis, item 13e, as to not overstate the strength of the literature. We calculated an average ICER where multiple studies existed within an EBI. We had no missing data (item 14) and used standard deviations to demonstrate dispersion around results (item 20d).
- Authors should provide a detailed account of the search strategy, potentially as a supplementary table, is necessary.
We have added this to the text and revised Figure 1 to follow clearly the PRISMA outline.
- Is the systematic review in PROSPERO? If not, discuss the limitations of this in the discussion section.
We were not eligible for registration in PROSPERO as we began our study search and data extraction prior to seeking registration. The aim of the register is to capture information at the design stage and we believe our paper reports our approach. We have no reason to believe lack of registration in PROSPERO contributed any bias to our work. We have, however, noted this as a limitation as registration would have been preferable.
- Figure 1 should adhere to the PRISMA flow diagram format, accurately depicting the articles retrieved from each database in distinct boxes, with the inclusion of the number of duplicated articles.
We have amended Figure 1 to conform to the PRISMA format.
Furthermore, the exclusion of 199 articles based solely on title screening necessitates careful consideration, and the abstracts of these studies should undergo thorough evaluation.
- Detailed inclusion and exclusion criteria, including factors such as sample size considerations, must be clearly outlined to enhance transparency and reproducibility. Additionally, clarification regarding the independence of article selection, variable extraction, and methodological quality assessment by two reviewers is imperative.
Per above we have added details on inclusion/exclusion criteria and text on sample size considerations. Note: We did not exclude any studies based on sample size as our goal was to include all identifiable EBI implementation studies. Also, implementation trials often have very small samples as they are conducted small community-based organizations and their focus on is hard-to-reach women. However, one study was excluded as the authors stated that attrition led to a small sample size that limited authors’ ability to accurately report ICERs.
All authors participated in article reviews. For initial elimination from the n=224 was based on reviews by two authors with reasons listed in the text. From that point forward all papers were assessed by all three authors.
Finally, the discussion section should provide a comprehensive justification of the practical implications of the study's findings. Without this, the manuscript risks being perceived as a mere summary of existing literature rather than a meaningful contribution to the field.
We have augmented the discussion to emphasize the following. 1) The literature at best provides very preliminary estimates of cost-effectiveness and declaring an EBI as cost-effective based on existing studies is not warranted. Strong evidence of effectiveness does not imply cost-effectiveness. 2) Advances in cost-effectiveness methodology are needed, particularly in the areas of sensitivity analysis, replicability, and generalizability. 3) It is of concern that microsimulations (not reviewed here) rely on assumptions about the increase in screening generated by outreach. Assumptions on this input need careful review. 4) Research on cervical cancer is lacking.
Reviewer 2 Report
Comments and Suggestions for Authors
In the systematic review examining the " Evaluation of the Cost Effectiveness of Evidence-based Interventions to Increase Female Breast and Cervical Cancer Screens," researchers have compiled comprehensive information regarding the typical expenses involved in preventive screenings for breast and cervical cancer. Their analysis reveals significant disparities in costs among various screening methods, as well as variations in their efficacy. Insights gained from such studies hold the potential to inform the development of economically viable screening strategies crucial for detecting and preventing cancer effectively.
Comments on the Quality of English LanguageThe manuscript's language is great for the scientific community, may be improved for the general population.
Author Response
Reviewer #2
In the systematic review examining the " Evaluation of the Cost Effectiveness of Evidence-based Interventions to Increase Female Breast and Cervical Cancer Screens," researchers have compiled comprehensive information regarding the typical expenses involved in preventive screenings for breast and cervical cancer. Their analysis reveals significant disparities in costs among various screening methods, as well as variations in their efficacy. Insights gained from such studies hold the potential to inform the development of economically viable screening strategies crucial for detecting and preventing cancer effectively.
Reviewer 3 Report
Comments and Suggestions for Authors
This is a very interesting paper that focuses on the evaluation of the cost-effectiveness of evidence -based interventions to increases female breast and cervical cancer screens via a systematic review. Overall it is a well written paper however, there are some questions and hopefully the authors will be able to answer those.
a) In the introduction section, need more information about the National Breast and Cervical Cancer Early Detection Program. The authors do mention that the program targets minority groups or underserved groups. But they do not include in the reviews any programs targeting African American women or Native American women. Why is that?
b) Regarding table 1, the authors use information from the community guide CPSTF. The information regarding mass media is outdated since the review was done in 2005. There is more recent review of evidence of improving HPV screening in Asia and especially Australia. Also, social media is also used in promoting HPV vaccination and screening. I think the authors need to include some more recent findings on the use of media, and health communications not only in the US but also globally.
c) Methods:
a. Missing reference on the definition of EBI cost-effectiveness.
b. Not clear what the inclusion criteria are. For example shouldn’t the trials first been identified as efficacious or effective before someone calculates the cost-effectiveness. An ineffective trial or intervention in terms of enhancing screening rates should be considered for cost-effectiveness. Why the effectiveness interventions were excluded?
c. There is a disagreement between the narrative and the information on Figure 1. For example, it is stated that the authors excluded 18 lit. review papers in the figure but in the narrative 19. Similarly, in the other outcome category, it is stated 50 in the figure and 5 in the narrative. Which one is right?
d. ICER is a ratio that includes not only the change in costs but also the change in effectiveness. The authors include only the amount of money ( cost of additional woman screened). Shouldn’t also include the change in the effectiveness? This is not clear.
e. Regarding the CHEERS clecklist, there are 28 items included. Why did the authors choose to show only some? What do the terms analytic horizon mean and that does the baseline effectiveness =0 or not 0 mean? Why look at baseline effectiveness and not only look at the long-term effectiveness? This is not clear to me.
D) Results:
Not clear how the ranking was done as the most cost-effective intervention based on the various parameters used based on percent change in screening rate, cost per additional women screening and ICER group average.
E). Discussion
The discussion seem in line with the results however, more emphasis should be put in the methodology and in describing in detail the different terms used because readers with no background in health economy will have a hard time understanding this paper. I would have liked more emphasis on lack of cost-effective interventions that take place among African American and Native American women. Why researchers or practitioners do not conduct cost-effective interventions targeting these two minority groups and do we need to expand the field of cost-effectiveness or cost-benefit analysis in this area? How will this information be used to improve the existing National Breast and Cervical Cancer Early Detection Program?
Author Response
Review #3
This is a very interesting paper that focuses on the evaluation of the cost-effectiveness of evidence -based interventions to increases female breast and cervical cancer screens via a systematic review. Overall it is a well written paper however, there are some questions and hopefully the authors will be able to answer those.
- In the introduction section, need more information about the National Breast and Cervical Cancer Early Detection Program. The authors do mention that the program targets minority groups or underserved groups. But they do not include in the reviews any programs targeting African American women or Native American women. Why is that?
Please note that we did not exclude any EBI cost-effectiveness reviews based on populations. Many outreach programs aim to serve hard-to-reach women, but a key point, which the reviewer raises below, is that only one focused on African American women. This is an important finding, now emphasized in the discussion per comments below at item E.
- Regarding table 1, the authors use information from the community guide CPSTF. The information regarding mass media is outdated since the review was done in 2005.
There is more recent review of evidence of improving HPV screening in Asia and especially Australia. Also, social media is also used in promoting HPV vaccination and screening. I think the authors need to include some more recent findings on the use of media, and health communications not only in the US but also globally.
The guide is generally updated every three to five years as more evidence becomes available. There is no evidence on mass media outreach for breast and cervical cancer. We have noted a review related to HPV per the reviewer’s suggestion.
- c)Methods:
- Missing reference on the definition of EBI cost-effectiveness.
Added reference.
- Not clear what the inclusion criteria are. For example shouldn’t the trials first been identified as efficacious or effective before someone calculates the cost-
We have expanded on the inclusion criteria under methods. We selected studies which implemented evidence-based interventions, classified as recommended or with insufficient evidence as established by the CPSTF. We included those with insufficient evidence as we want to include the widest array of interventions and add to the overall knowledge base for CPSTF. We have distinguished this category more clearly in the text. We did not include interventions which were not recommended.
An ineffective trial or intervention in terms of enhancing screening rates should be considered for cost-effectiveness. Why the effectiveness interventions were excluded?
Response: We are not clear on the reviewer’s question, but attempted to answer it below.
Two trials proved to be ineffective when they implemented an intervention identified by the Community Guide. These studies show that implementation of an evidence-based intervention is not always efficacious in a different setting, an important finding and underscores the importance of assessing the replicability of interventions before widespread adoption.
Also, in some studies, cost and effectiveness evaluation are evaluated simultaneously, so establishing effectiveness a prior is not a study goal, rather the authors are implementing their interventions based on CPTSF guidelines.
- There is a disagreement between the narrative and the information on Figure 1. For example, it is stated that the authors excluded 18 lit. review papers in the figure but in the narrative 19. Similarly, in the other outcome category, it is stated 50 in the figure and 5 in the narrative. Which one is right?
Corrected these discrepancies. Thank you for catching this.
- ICER is a ratio that includes not only the change in costs but also the change in effectiveness. The authors include only the amount of money ( cost of additional woman screened). Shouldn’t also include the change in the effectiveness? This is not clear.
The reviewer is correct and we have clarified this point. As we indicated, the ICER is calculated as the cost per additional woman screened (InterventionCost - StatusQuo Costt/InterventionScreeningRate – StatusQuoScreeningRate) We state this as cost per additional woman screened. In actuality, we are calculating the net cost(savings) per woman screened. The convention in the literature, however, is to omit the word, net.
- Regarding the CHEERS clecklist, there are 28 items included. Why did the authors choose to show only some?
We summarized the most important aspects of the studies in the table for practical presentation and addressed others in the results text and/or discussion. Some items were not applicable, such as details on modelling as no study used modelling. We now note this in the text.
What do the terms analytic horizon mean and that does the baseline effectiveness =0 or not 0 mean? Why look at baseline effectiveness and not only look at the long-term effectiveness? This is not clear to me.
We have added notes to Table 2 to define these terms. The term analytic horizon is the period of time over which outcomes were evaluated. For example, here the analytic horizon is the period over which women were observed to see if they obtained a screen after participating in outreach.
Baseline effectiveness is the “status quo” effectiveness or the rate of screening prior to implementation of the outreach program. It is used in the ICER denominator (noted above).
- D) Results:
Not clear how the ranking was done as the most cost-effective intervention based on the various parameters used based on percent change in screening rate, cost per additional women screening and ICER group average.
We have clarified this in the text and included the following. We summarize the ICER value for each EBI and present an average ICER for EBIs with multiple studies in the tat category. We rank EBIs in the table from most cost-effective (the lowest ICER) to least cost-effective (the highest ICER. As the goal of all the programs is to increase screening rates, we want to identify the EBI program whereby an additional woman can be screened at lowest cost.
E). Discussion
The discussion seem in line with the results however, more emphasis should be put in the methodology and in describing in detail the different terms used because readers with no background in health economy will have a hard time understanding this paper.
We have aimed to clarify language per the above and added text regarding methodology in the limitations section.
I would have liked more emphasis on lack of cost-effective interventions that take place among African American and Native American women. Why researchers or practitioners do not conduct cost-effective interventions targeting these two minority groups and do we need to expand the field of cost-effectiveness or cost-benefit analysis in this area?
We appreciate the reviewer raising this point and agree that it needs more emphasis. The challenge here is many outreach programs may have intended to increase screening rates among African American and Native American women, the result is that they are not reaching these populations. Thus, other forms of interventions, places of outreach need to be considered. Also, Native American women present a specific challenge as tribes operate self-contained health care systems and reaching this group for additional, outside care for which they are eligible, is very difficult.
How will this information be used to improve the existing National Breast and Cervical Cancer Early Detection Program?
The NBCCEDP is increasingly interested in cost-effectiveness studies as a potential mode of allocating program support dollars. The main message is that the program should exercise caution in using cost-effectiveness rankings as the basis for resource allocation. Current estimates of cost-effectiveness across and within EBIs are highly variable and methodological issues remain, for example in terms of sensitivity analysis, sample size estimation etc. A potentially important area for NBCCEDP to fund cost-effectiveness studies addressing current shortcomings which can be used to provide input into prioritization, but the literature does not offer that at this point.
Round 2
Reviewer 1 Report
Comments and Suggestions for Authors
Authors have included all the relevant comments.